# Angiographic Outcomes for Arterial and Venous Conduits Used in CABG

**DOI:** 10.3390/jcm12052022

**Published:** 2023-03-03

**Authors:** Arnaldo Dimagli, Giovanni Soletti, Lamia Harik, Roberto Perezgrovas Olaria, Gianmarco Cancelli, Kevin R. An, Talal Alzghari, Charles Mack, Mario Gaudino

**Affiliations:** 1Department of Cardiothoracic Surgery, Weill Cornell Medicine, New York, NY 10065, USA; 2Department of Cardiothoracic Surgery, New York Presbyterian Queens Hospital, Queens, New York, NY 11355, USA

**Keywords:** coronary artery bypass grafting, arterial grafts, radial artery, thoracic artery, saphenous vein, total arterial grafting

## Abstract

Coronary artery bypass grafting is the most commonly performed cardiac surgical procedure. Conduit selection is crucial to achieving early optimal outcomes, with graft patency being likely the main driver to long-term survival. We present a review of current evidence on the patency of arterial and venous bypass conduits and of differences in angiographic outcomes.

## 1. Introduction

Coronary artery bypass grafting (CABG) is the most commonly performed cardiac surgery and the treatment of choice in patients with complex and multivessel coronary artery disease (CAD) [1,2]. However, there is still discussion regarding optimal strategies for CABG, with the choice of conduit being a particularly highly debated topic.

The long-term survival benefit provided by CABG likely relies on the long-term patency of the grafts utilized during the procedure. Arterial grafts have been suggested to provide better long-term patency by being less susceptible to atherosclerotic disease and, therefore, occlusion compared to venous grafts [3]. However, current national registries in the United States, United Kingdom, and globally report low use of arterial grafts in patients undergoing CABG [4,5,6].

The aim of this review is to provide a state-of-the-art summary of the current evidence on the patency of grafts used in CABG.

## 2. Saphenous Vein

The saphenous vein graft (SVG) was first used by Favaloro in 1968 [7] and is currently the most commonly used conduit for revascularization. This is due to its ease of availability, its accessibility, and its length. While the left internal thoracic artery (LITA) remains the first choice of conduit in CABG, the high usage rate of the SVG may also be related to the lack of consensus on either the radial artery (RA) or the right internal thoracic artery (RITA) as the second conduit of choice in CABG.

Graft failure represents the key problem with SVG use: 10–25% of SVGs fail within the first year after CABG, another 5–10% fail between one and five years after CABG, and up to 25% fail in the following years, leading to an overall failure rate of almost 50% at 10 years [8,9,10,11]. Early graft failure is generally related to technical issues and graft thrombosis [8]. The late failure phase is instead ascribed to the development of disease intrinsic to the graft, such as intimal hyperplasia and progressing atherosclerosis [8]. Several additional factors can impact the short- and long-term patency of SVGs.

### 2.1. The Impact of the Harvesting Technique

The observation that injury and stress of the endothelium, which is incurred intraoperatively during graft harvesting, could be behind the suboptimal patency of SVGs prompted the development of new harvesting approaches. In the 1990s, the no-touch (NT) technique was first described: the SVG is atraumatically harvested with the surrounding pedicle [12], which prevents graft spasm and retains the biological properties of the vein, such as the production of nitric oxide [13]. In contrast to the conventional technique, manual dilatation of the SVG is not needed with NT harvesting, sparing the SVG of mechanical endothelial injuries [12,14]. Finally, the surrounding adipose-connective tissue can act as a supporting scaffold that prevents the graft from kinking.

A single-centered randomized clinical trial investigated the patency rate of SVGs in 156 patients randomized to conventional SVG harvesting with conduit distension, conventional SVG harvesting without conduit distension, or NT harvesting. The study consistently demonstrated at 1.5, 8.5, and 16 years that NT-SVG had the lowest rate of graft failures, with patency (83%) comparable to that of the internal thoracic artery (ITA) [15,16,17].

Another small, randomized trial, the Impact of Perivascular Tissue on Endothelial Function in Coronary Artery Bypass Grafting (IMPROVE-CABG), randomized 100 patients undergoing CABG to either conventional or NT-SVG harvesting. The early results showed that the occlusion rates were not different (conventional 1.2% vs. NT 0.8%), and the prevalence of harvest site infection was comparable in the two groups (4%) [18]. The five-year angiographic results of this trial are anticipated.

The SUrgical and Pharmacological novel intERventions to Improve Overall Results of Saphenous Vein Graft Patency (SUPERIOR SVG) trial was the first multicenter angiographic randomized clinical trial comparing the conventional versus the NT technique in 250 patients undergoing CABG in 12 centers [19]. The incidence of the primary composite outcome of SVG occlusion and all-cause mortality one year after surgery was not different between the two techniques (conventional 10.6% vs. NT 5.5%; odds ratio [OR] 0.49, 95% confidence interval [CI] 0.19–1.28). Similarly, there was no difference in the secondary composite outcome of significant stenosis or complete occlusion (conventional 15% vs. NT 7.8%; OR 0.48, 95% CI 0.20–1.19). Importantly, the incidence of leg (harvest site) adverse events, defined as infection, necrosis, dehiscence, drainage, and fluid collection, were higher in the NT group (NT 23.3% vs. conventional 9.5%; *p* < 0.01). However, the cumulative incidence of wound infection became comparable at one year [19]. When the results of these three trials were pooled meta-analytically, the estimate for the composite of graft occlusion and significant stenosis favored the NT-SVG harvesting technique (OR 0.47, 95% CI 0.26–0.84) [19].

The Graft Patency Between no-touch Vein Harvesting Technique and Conventional Approach in Coronary Artery Bypass Graft Surgery (PATENCY) trial randomized 2655 patients undergoing CABG in seven centers in China to either NT or conventional SVG harvesting [20]. The odds of vein graft occlusion at 12 months were found to be decreased with the NT harvesting technique (6.5% vs. 3.7%; OR 0.56, 95% CI 0.41–0.77). Moreover, recurrence of angina was significantly lower in the NT group, while the rates of major adverse cardiac and cerebrovascular events were not different between the two groups (conventional 4.3% vs. NT 3.8%; OR 0.89, 95% CI 0.61–1.29). Before discharge, patients in the NT group were more likely to develop exudation, numbness, or edema at the harvest site, but the rate of wound re-suture was not different compared to the conventional harvesting approach. At one year follow-up, there was no difference in terms of wound pain and wound surgical intervention between the groups. An important caveat of the study is that vein harvesting was performed by senior residents who had performed at least 100 SVG harvests and had undergone standardized training for NT technique prior to trial enrollment [20].

Further evidence will be available after the publication of the results of the ongoing SWEDEGRAFT trial, which is investigating the two-year patency results of the NT and conventional SVG harvesting techniques, as well as the two-year rates of leg wound complications [21].

Endoscopic vein harvesting (EVH) was introduced to decrease the incidence of leg wound complications. In a meta-analysis of 22 studies, including 27,911 patients, harvest-site complications were less common in the EVH group compared to the open harvesting group (0.75% vs. 2.92%; OR 0.19, 95% CI 0.12–0.30; *p* < 0.001) [22]. There was also no difference in the incidence of all-cause mortality and major adverse cardiac events between open and EVH, despite the finding that the EVH was associated with a lower SVG patency rate at one to five years (OR 0.80; 95% CI 0.70–0.91) and at 5–7 years (OR 0.15, 95% CI 0.04–0.61).

### 2.2. The Impact of Pharmacotherapy

The use of antiplatelet agents (aspirin and P2Y12-inhibitors) and statins are aimed at preventing thrombosis within the graft secondary to the activation of the endothelium and at preventing the development of intimal hyperplasia and atherosclerosis within the graft, respectively. The use of these medications is the cornerstone of medical management after CABG in order to maintain graft patency.

In a network meta-analysis including 16 studies and 3133 patients, the use of aspirin compared to placebo was associated with a significant improvement in SVG patency (OR 1.90, 95% credible interval: 1.3–2.8). A similar favoring trend, albeit not a significant difference, was shown when aspirin alone was compared to aspirin plus clopidogrel or ticagrelor [23]. However, the meta-analysis did not formally investigate clinical outcomes, such as bleeding and cerebrovascular accidents. It is reasonable to hypothesize that there is an enhanced antiplatelet effect with the concomitant use of aspirin and ticagrelor or clopidogrel (DAPT, dual antiplatelet therapy). However, studies comparing aspirin alone and DAPT reported conflicting results [24,25,26].

The Different Antiplatelet Therapy Strategy after CABG Surgery (DACAB) was a multicenter, randomized clinical trial that enrolled 500 patients assigned to either aspirin alone, ticagrelor alone, or ticagrelor DAPT within 24 h from CABG. At one year, there was a significant difference in SVG patency between aspirin alone (76.5%) and ticagrelor DAPT (88.7%; difference 12.2%, 95% CI 5.3–19.2%; *p* = 0.01), while there was no difference between aspiring alone and ticagrelor alone (82.8%; difference 6.3%, 95% CI −1.1–13.7; P0.10). Notably, there was no difference in the rate of major bleeding between the three groups [25]. Different results were demonstrated in the POPular CABG trial (The Effect of Ticagrelor on Saphenous Vein Graft Patency in Patients Undergoing Coronary Artery Bypass Grafting Surgery), which enrolled 499 patients randomized to receive either ticagrelor or placebo in addition to standard aspirin therapy. The incidence of SVG occlusion was 10.5% in the ticagrelor DAPT group and 9.1% in the placebo group, leading to a non-significant treatment effect (OR 1.29, 95% CI 0.73–2.30). The rate of major bleeding was not different between the two groups (2.0% vs. 2.0%; *p* > 0.99) [26].

In the recently published Ticagrelor Antiplatelet therapy to Reduce Graft Events and Thrombosis (TARGET) trial, 250 patients were randomly assigned to receive either aspirin or ticagrelor after CABG. At two years after surgery, the angiographic follow-up demonstrated that neither SVG occlusion (aspirin 15.7% vs. ticagrelor 13.2%; *p* = 0.71) nor the presence of any SVG disease (aspirin 19.4% vs. ticagrelor 19.8%; *p* = 1.00) were different between the two groups. Similar to previous studies, the incidence of bleeding was not different between aspirin alone (2.3%) and ticagrelor alone (5.8%) [27].

All these trials were limited by the small sample size and by being underpowered for the detection of relevant clinical outcomes, such as bleeding. The use of an individual patient-data meta-analysis merging those studies was eventually able to provide more solid results. A total of 1316 patients and 1668 SVGs were included and compared the SVG failure rate between aspirin alone, ticagrelor alone and ticagrelor DAPT [28]. The authors found that ticagrelor DAPT was associated with a risk of SVG failure that was nearly half that of aspirin, a significant difference (OR 0.51, 95% CI 0.35–0.74; *p*  <  0.001), but that ticagrelor DAPT was also associated with an increased risk of bleeding when compared with aspirin (OR 2.98; 95% CI 1.99–4.47; *p*  <  0.001). Ticagrelor alone was not associated with SVG failure or a higher incidence of bleeding. Therefore, the authors concluded that the risk of failure of the SVG should be balanced carefully within each patient, with the risk of bleeding arising from DAPT [28].

Current North American guidelines recommend initiating aspirin within six hours postoperatively and continuing indefinitely as a strategy to reduce SVG occlusion and averse cardiovascular death (Class I, level of evidence [LOE] A) [1]. In selected patients, such as those undergoing off-pump CABG, DAPT should be considered (Class 2b, LOE B-R) to improve vein graft patency as compared to aspirin alone [1].

Statins have been proven to reduce the incidence of SVG occlusion through their demonstrated cholesterol-lowering effect [29,30,31].

A small, single-center angiographic study showed that patients who received statins preoperatively had a significantly reduced number of new lesions and a lower incidence of SVG occlusion (*p* = 0.02) one year after CABG, which translated into a significant decrease in the incidence of myocardial infarction (MI) [31].

The widespread habit of initiating long-term statin therapy after bypass surgery was validated in a retrospective study including 7503 patients who underwent CABG between 1995 and 2004. Compared to no statin use, the commencement of statin therapy within one month postoperatively was associated with lower all-cause mortality (HR 0.82, 95% CI 0.72–0.94) and a lower risk of major adverse cardiovascular events (HR 0.89, 95% CI 0.81–0.98) [32]. These findings were confirmed in a retrospective analysis including 35,193 patients who underwent first-time isolated CABG in Sweden from 2006 to 2017. The benefit of initiating a statin therapy also translated into a markedly reduced risk of undergoing new angiography (HR 0.81, 95% CI 0.74–0.88) and new revascularization (HR 0.79, 95% CI 0.70–0.88) [33].

Lastly, while no doubt remains regarding the efficacy of statins in reducing SVG occlusion, debate still surrounds the dose to obtain this benefit. The Aggressive Cholesterol Therapy to Inhibit Vein Graft Events (ACTIVE) trial showed that an 80 mg dose did not confer a significant advantage in terms of vein graft occlusion one year after CABG compared to a 10 mg dose (11.4% for 80 mg versus 12.9% for 10 mg) [34].

Current North American guidelines recommend all patients undergoing CABG should receive statin therapy prior to and early after surgery, to be indefinitely continued unless contraindicated [1]. Dose intensity may be adjusted based on the patient’s age, LDL target, statin intolerance status, and the presence of drug interactions [1].

### 2.3. The Impact of External Stenting

SVG failure after one year following CABG is mainly due to intimal hyperplasia, which predisposes the graft to accelerated atherosclerosis [8]. Attempts to attenuate this pathophysiologic mechanism have led to the emergence of a novel surgical strategy based on the use of cobalt-chromium mesh external stents that have been shown to be promising in preclinical testing by reducing wall tension, improving lumen uniformity, and creating a protective “neo adventitia” layer rich with microvasculature [5,35,36].

First experiences with such devices required gluing, suturing or both during the implantation procedure and yielded high rates of graft occlusion at up to one-year follow-up [37,38,39,40]. This initial failure prompted the industry to improve the design of external stents and led to the conduction of the Venous External Support Trials (VEST). In the VEST I trial, 30 patients were enrolled and received one stented SVG (30 grafts), and one or more non-stented SVG (39 grafts) [41]. The intimal hyperplasia area, assessed by intravascular ultrasonography, was lower in the stented group (mean 4.37 mm^2^) compared to nonstented grafts (mean 5.12 mm^2^; *p* = 0.04). The one-year angiographic follow-up, completed in 97% of patients, showed similar failure rates between stented and non-stented grafts (30% vs. 28.2%; *p* = 0.55). The five-year follow-up (VEST IV) confirmed similar patency rates between the two groups (30% vs. 23%; *p* = 0.42), while the intimal hyperplasia area was still significantly smaller in stented SVGs (*p* < 0.001) [42].

The VEST II trial investigated factors that could impact the patency of stented SVG supplying the right coronary territory finding that the use of only suture ligation and avoidance of fixation of stents to the proximal or distal anastomosis led to similar patency rates for stented and non-stented grafts (86.2% vs. 88.8%) at 3–6 months [43].

The VEST III trial enrolled a larger cohort of patients (*n* = 184) and showed similar patency rates between stented and non-stented SVGs (78.3% vs. 82.2%; *p* = 0.43) at two years [44]. Therefore, based on the available evidence, external stenting certainly plays a relevant role in preventing the development and progression of graft atherosclerosis. However, further long-term data and device refinement are warranted to improve upon this promising technique, as angiographic outcomes seem to be in favor of non-stented SVGs.

### 2.4. The Impact of the Anastomotic Technique

SVGs can be grafted as an individual anastomosis or as a sequential anastomosis, which allows for a greater extent of myocardial revascularization. However, there is limited data on the impact of sequential grafting on SVG patency.

In a study of 2515 patients, the clinical and angiographic outcomes in 946 sequential SVGs and 1366 individual SVGs were compared [45]. The graft failure rates were 10.3% and 17.7% at five years and 20.9% and 33.6% at 10 years in the sequential and individual groups, respectively. After propensity score adjustment, the risk of graft failure was lower in the sequential group at a median follow-up of 88 months (HR 0.69, 95% CI 0.50–0.95; *p* = 0.02). The risk of the composite outcome of death, nonfatal MI, and repeat revascularization was numerically lower in the sequential group (36.8% vs. 41.4%; HR 0.91, 95% CI 0.75–1.09; *p* = 0.30). Conversely, in a sub-study of the Project of Ex-Vivo Vein Graft Engineering via Transfection (PREVENT) IV trial, which enrolled 3014 patients to receive either edifoligide-treated SVGs or placebo, investigators found a higher risk of graft occlusion with sequential anastomosis at one year (OR 1.24, 95% CI 1.03–1.48; *p* = 0.025) [46].

A recent meta-analysis including 15 cohort studies, totaling 10,681 patients and 12,957 grafts found that patency was higher for sequential SVGs compared to individual SVGs (relative risk [RR] 1.11, 95% CI 1.03–1.21; *p* = 0.01) [47].

Due to conflicting results, more solid evidence, potentially from RCTs, is welcomed to explore the impact of individual vs. sequential anastomosis on SVGs patency.

## 3. Internal Thoracic Artery

The use of the LITA has been the gold standard for surgical myocardial revascularization since Loop et al.’s study demonstrating a long-term benefit in terms of mortality, MI, and cardiac reintervention in patients receiving a single ITA to the left anterior descending coronary artery (LAD) compared with patients receiving only venous grafts [48]. Further studies have later corroborated this evidence, in particular supporting the hypothesis of the underlying mechanism for better long-term results: better graft patency [49,50]. The use of LITA for revascularization of the LAD is now recommended by international guidelines on myocardial revascularization due to its exceptional durability [1,2]. Long-term patency rates for the LITA are outstanding: 93% at 10 years and 88% at 15 years in the latest reports [51,52].

The right internal thoracic artery (RITA) has not achieved a similar widespread success compared to its contralateral counterpart, and its use is still mainly concentrated in the hands of surgeons who routinely perform bilateral internal thoracic artery (BITA) grafting. The long-term patency rates for the RITA are reported to be between 80–90% at 10 years [53,54].

There may be several explanations for the failure of BITA to achieve widespread adoption, but the neutrality of the Arterial Revascularization Trial (ART) played a pivotal role. The ART was a multicenter, double-arm study investigating the outcomes of 3102 patients undergoing CABG and randomized to either single internal thoracic artery (SITA) or BITA grafting. At the ten-year follow-up, the intention-to-treat analysis showed no difference in all-cause mortality and event-free survival between the two groups. After the publication of this trial, important issues within the trial were highlighted. First, the crossover rate in the two groups was notably high, with 14% of patients in the BITA group actually receiving SITA, diluting the treatment effect of the BITA. Second, there was no a priori criteria to guide surgeons in the choice of concomitant arterial conduits, leading to 21% of the patients in the SITA group receiving an RA as well. In the as-treated analyses, there was a significantly lower risk of mortality in the BITA group. However, these findings can only be considered hypothesis-generating.

The results from ART contrasted substantially with the findings from observational studies, which had identified a benefit from the use of the two ITAs [55]. Among other factors impacting the overall adoption of the BITA are the higher complexity and longer times required for harvest [56] and the perceived higher risk of sternal wound complications. Prior reports have also highlighted the dependency of BITA outcomes on operator experience [57], possibly limiting adoption among surgeons who fear they lack the requisite expertise. The combination of these issues has led to limited use of BITA: around 5% in the U.S. [4] and 15% in Europe [58].

Current North American guidelines indicate that BITA should be used by experienced surgeons in appropriately selected patients (Class 2a, LOE B-NR) [1], which is aligned with current European recommendations [2].

### 3.1. The Impact of the Harvesting Technique

The ITA can be harvested either as a pedicled or as a skeletonized graft. Traditionally, this conduit has been harvested as a pedicled graft, meaning that the accompanying veins, the endothoracic fascia, and parts of the parietal pleura and of the transversus thoracic muscle are taken together with the artery. Conversely, skeletonized ITA entails the dissection of the artery as an isolated conduit free from the aforementioned surrounding tissues. Skeletonization is more challenging and technically demanding, with a steep learning curve and longer harvesting time (15–20 min compared to pedicled harvesting) [59,60]. In addition, the need to dissect more closely to the ITA wall can increase the risk of injuries induced by electrocautery or manipulation of the vessel (i.e., hematoma, vessel dissection, thrombosis), which can impact graft patency.

These two techniques were recently at the center of the debate in the cardiac-surgical community following the publication of two post-hoc analyses utilizing data from randomized clinical trials.

The Cardiovascular Outcomes for People using Anticoagulation Strategies (COMPASS) clinical trial originally randomized 27,395 patients with stable CAD or peripheral artery disease to receive rivaroxaban and aspirin or rivaroxaban and placebo or aspirin and placebo for secondary cardiovascular prevention [61]. Of those, 1002 patients undergoing CABG were included in this analysis [62]. The ITA was most commonly harvested as a pedicle graft (72%). After a mean follow-up of 23 months, patients receiving a pedicled ITA had a higher risk of major adverse cardiovascular events (cardiovascular mortality, MI, stroke, or revascularization; HR 3.19, 95% CI 1.53–6.67). This difference was mainly driven by the higher rate of revascularization in the skeletonized group (5% vs. 1.4%; *p* = 0.03). With regards to the graft status, skeletonized ITAs had a significantly higher incidence of occlusion (9.6% vs. 3.9%; OR 2.41, 95% CI 1.39–4.20; *p* = 0.002). These findings were confirmed when the quality of the conduit and the quality of the target coronary vessel were taken into account. It is important to note that the original randomized treatments were not found to have an impact on graft patency.

The ART [63] investigators conducted a post-hoc analysis of 2161 patients, of whom 58% received a pedicled ITA [64]. At the 10-year follow-up, patients receiving skeletonized ITAs were found to have a higher risk of major adverse cardiovascular outcomes (all-cause mortality, MI, and revascularization; HR 1.25, 95% CI 1.06–1.47) [64]. In particular, patients with skeletonized ITA experienced a higher rate of repeat revascularization (13.5% vs. 9.9%; *p* = 0.01). The findings were confirmed when only patients receiving single ITA were included in the analysis, eliminating the potential bias arising from the greater likelihood of using the skeletonization technique with BITA. The post-hoc analysis of the ART also attempted to account for the surgeons’ experience. When the surgeon volume was ≥51 cases, the differences in outcomes between skeletonized and pedicled IMAs were no longer present. A major limitation was the lack of angiographic data.

Both these post-hoc analyses raise questions on the use of ITA as a skeletonized graft. First, skeletonization may impair the physiological endothelial function of the ITA by direct damage to the vasa vasorum of the vessel [65,66]. Second, from a more careful assessment of the Kaplan–Meier curves, it is possible to see that the survival curves diverge early on within the first 12 months after surgery [62,64]. This observation could support the hypothesis that technical issues rather than pure advancement of the atherosclerotic disease are the causes for the poorer performance of the skeletonized ITA. Attention is thus directed to the surgeon and the surgeon’s experience as a limiting step in the successful performance and adoption of the skeletonization of the ITA. In fact, in the post-hoc analysis of the ART, when surgeons had a higher surgical volume, the outcomes between the two techniques were not different.

Finally, a recent randomized trial investigating graft patency of 109 patients undergoing CABG with either skeletonized or pedicled ITA grafted to the LAD showed no difference in graft patency at three years (pedicled 95.8% vs. skeletonized 90.4%; computed angiograms available for 100 patients) or at eight years (pedicled 93% vs. skeletonized 90.2%; computed angiograms available for 84 patients) [67]. The authors showed that target vessel stenosis (defined as stenosis of <70% vs. ≥70%) was a more important factor for graft failure rather than the harvesting technique.

### 3.2. The Impact of Graft Configuration and Target Coronary Vessel

The RITA graft can be used as an in-situ graft or as a free graft coming off of the aorta, the LITA as a Y-composite conduit, or the aortic hood of the SVG or RA (Figure 1).

In a randomized control trial of 304 patients, of whom 147 were assigned to in-situ RITA and 152 to RITA Y-graft, there was no difference in patency between the two configurations at six months. The observed anastomotic patency rate was 97% in both groups [68]. No differences in clinical outcomes in terms of major adverse cerebral and cardiovascular events were reported. The extended follow-up of this cohort, including 75% of the original population, confirmed no difference in the patency rate of in-situ RITA (93%) and Y-graft (94.5%) at a mean angiographic follow-up of three years. However, the incidence of major cerebral and cardiovascular events was higher with the in-situ configuration (34% vs. 25%) at a mean clinical follow-up of seven years [69]. Similar findings were reported in a study of 1818 patients undergoing CABG with either in-situ RITA or Y-graft RITA. The patency rate was 100% for the in-situ grafts and 99.2% for the Y grafts at a mean follow-up of 17 months [70].

In a recent study of 1331 patients from the Cleveland Clinic, the patency rate of the in-situ RITA (91%) was similar to the patency rate of the RITA as a free graft from the aorta (91%) or from the hood of the SVG (77%), and as a Y-graft (89%) [71]. The authors showed that the RITA patency rate was independent of inflow configuration but was significantly influenced by the target coronary vessel. In-situ RITA to LAD had similar patency to LITA to LAD (94.4% vs. 95.4%; *p* = 0.50), while a lower patency rate was found when in-situ RITA was used to graft non-LAD territories [71]. Similar results were described in a study of 5776 patients undergoing BITA-CABG, showing no difference in the patency of in-situ vs. free RITA grafts (89% vs. 91%) at ten years but the patency of RITA grafts to the right coronary artery was significantly lower compared to RITA to LAD grafts [72]. The authors also identified target coronary vessel stenosis as a factor influencing RITA patency, having observed that RITA grafts anastomosed to vessels with <60% stenosis had the lowest patency rate and grafts anastomose to vessels with >80% stenosis had the highest patency rate [72].

### 3.3. The Impact of the Anastomotic Technique

The use of sequential anastomosis for ITA can be key to achieving effective revascularization in patients with a limited number of grafts and diffuse CAD (Figure 2).

Current evidence, albeit based on observational studies, seems to be concordant with sequential ITA having a similar patency rate as individual ITA [73,74,75,76]. A recent study including 120 propensity score matched pairs reported that the patency rate of the sequential LITA graft to the diagonal artery and LAD was similar to the patency of individual LITA-to-LAD grafts (99% and 98% vs. 98%; *p* > 0.05) at the angiographic follow-up of 27 months. [76]. Similar rates were found in a report of 101 patients in which patency rates for the sequential LITA to the circumflex and right coronary artery were 98% and 95%, respectively [74].

Interestingly, the design of the sequential anastomosis was reported to impact graft patency [75]. In sequential ITA grafting, the anastomosis can be performed either in a “diamond” fashion with the conduit perpendicular to the coronary target or in a parallel fashion with the conduit parallel to the coronary target. A study of 452 patients comparing LITA sequential vs. individual grafting to the circumflex showed that a “diamond” anastomosis had a numerically higher patency rate of the distal segment of the sequential graft as compared to a parallel design (98.4% vs. 90.7%; *p* = 0.09). In particular, the lowest patency rate (75%) was reported when the proximal anastomosis of a sequential graft to the diagonal and circumflex arteries was placed in a parallel fashion [75].

However, it is important to note that the “diamond” design is more technically demanding and requires accurate precision of the arteriotomies in order to prevent a “seagull effect”, which can lead to a flattening of the graft and impairment of graft status.

Another important factor potentially affecting sequential graft success is coronary target stenosis. Lower stenosis can result in higher competitive flow and therefore jeopardize the functionality of the graft [77]. In the Impact of Preoperative FFR on Arterial Bypass Graft Function (IMPAG) trial, which included a total of 64 patients (199 anastomoses, of whom 108 sequential), a preoperative fractional flow reserve (FFR) of at least 0.78 was associated with a better graft functionality at the six-month angiographic assessment. Lower values were associated with an anastomotic occlusion rate of 3% [78]. In a post hoc analysis of the IMPAG trial, specifically looking at sequential grafts, cut-offs of 0.80 and 0.78 were found for the first and second anastomoses of sequential grafts to the anterior circulation compared to 0.74 for the individual grafts. The FFR cut-off was higher for grafts to the postero-lateral territory being 0.81 and 0.78 for the first and second anastomoses of sequential grafts and 0.79 for individual grafts [79]. These findings suggest that attention should be paid to the severity of the target coronary vessel when sequential anastomosis is planned.

## 4. Radial Artery

The radial artery (RA) was first used as a conduit in 1972 by Carpentier [80]. However, due to the high tendency to vasospasm, which led to poor outcomes, the early experience with this conduit was far from promising and its use was shortly abandoned. At two years after CABG, over one-third of the RA grafts demonstrated angiographic narrowing or occlusion [81].

In the last three decades, the use of the RA has been revived due to the interest in multi-arterial coronary grafting. This was paralleled by the report of excellent 18-year angiographic outcomes [82] and by the understanding that injury to the graft during the harvesting of the conduit, as opposed to the graft itself, was the cause of the previously reported early unsatisfying outcomes. Moreover, the use of vasodilators intraoperatively and postoperatively further improved outcomes by addressing the tendency of RA grafts to spasm.to spams of the graft [83,84].

The RA is an accessible and easily procured conduit that can provide enough length to reach most coronary targets (Figure 3).

Its diameter is proximally optimal for anastomosis to the aorta or to the LITA and distally excellent for coronary artery targets [85]. On the other hand, the RA is more muscular than the other arterial conduits and thus carries an increased risk of vasospasm during harvesting.

### 4.1. The Impact of Target Coronary Artery and Competitive Flow

The target coronary artery is one of the most important determinants of graft patency. Target vessel considerations that influence the graft selection process are myocardial territory to be grafted, degree of proximal coronary artery occlusion, and diameter of the target vessel. The RA, as discussed, is prone to spasm and competitive flow when grafted to coronary target vessels with mild-to-moderate stenosis [86]. A study including 123 patients undergoing CABG with RA and a total of 382 distal anastomoses reported an overall patency rate of 92% at a mean angiographic follow-up of 32 months [87]. Interestingly, authors found important modifiers of RA patency. When the RA was used to graft the right coronary system, the patency rate was significantly lower when the RA was grafted to the left coronary system (79% vs. 94%; *p* < 0.05). Also, RA distal anastomoses to coronary vessels with >90% stenosis showed a higher patency rate compared to RA distal anastomoses to coronary vessels with 50–90% stenosis (98% vs. 83%; *p* < 0.05) [87]. The pivotal role of target vessel stenosis was also confirmed in an analysis of 100 CABG patients receiving RA grafts; RA grafted to vessels with ≥90% stenosis had a failure rate of 13%, which was similar to the failure rate of the LITA. The failure rate was as high as 80% in grafts anastomosed to vessels with <90% stenosis [88].

Based on this evidence, current guidelines recommend the use of the RA as the second conduit to graft significantly stenosed coronary targets and the postoperative administration of calcium-channel blockers for the following year to prevent graft spasm [1,2].

### 4.2. The Impact of Sequential Grafting

Sequential grafting is often used to increase the number of territories revascularized by a single conduit. However, data on the patency of single versus sequential anastomosis is limited and controversial. A study of 410 CABG patients receiving RA grafting showed that sequential anastomoses of the RA were associated with a higher likelihood of patency as compared to a single anastomosis at mean angiographic follow-up of five years (HR: 2.53, 95% CI: 1.29–5.28; *p* = 0.006) [89]. Conversely, a recent study including 208 patients (293 anastomosis total) undergoing CABG with skeletonized RA found a similar patency rate for sequential grafts (88.7%) and single grafts (87.4%; *p* = 0.88) at ten years [90]. There is a potential impact of target vessel stenosis on RA sequential graft patency. A study including 432 patients and 1221 distal RA sequential anastomoses showed that patency rates were higher for sequential anastomoses to target vessels with 76–100% stenosis compared to sequential anastomoses to target vessel with a stenosis of 51–75% (88.6% vs. 59.1%; *p* < 0.001) [91].

### 4.3. The Impact of Graft Configuration

The RA can be used as a graft directly anastomosed to the proximal aorta or as a composite conduit with other grafts, such as the ITA (Figure 4).

The available evidence seems to suggest that inflow configuration does not affect the long-term patency rate of the RA. At a mean follow-up of 6.5 years, the patency of aorta-anastomosed RA was 92%, compared with a patency rate of 86.3% for ITA-anastomosed graft (*p* = 0.81) [92]. However, when the RA was used to bypass target vessels with <90% stenosis, ITA-anastomosed RA grafts were associated with a higher risk of failure (*p* = 0.02). The impact of trans-radial catheterization.

RA grafts previously used for heart catheterization may have lower patency rates compared to those that have not been catheterized grafts [93]. Injury to the RA by the introduction and passing of catheters and wires likely produces endothelial damage to the RA lumen, especially in the proximal RA [94,95]. The patency rate can be up to 20% lower in instrumented RA compared to never-catheterized arteries [93]. This observation was also associated with the findings that the incidence of intimal hyperplasia was 30% greater in previously catheterized RA [93]. As a consequence, previously instrumented RAs should be used with caution in coronary surgery [96].

## 5. Gastroepiploic Artery

The gastroepiploic artery (GEA) was first used by Bailey in 1967 [97] to revascularize the posterior ischemic myocardium: however, its use as a conduit was never widely adopted and remained a grafting option only in a minority of cases. The GEA is well-suited for coronary targets on the posteroinferior aspect of the heart, such as the posterior descending artery, where it can be anastomosed as an in-situ graft. Harvesting of the GEA requires an extension of the median sternotomy beyond the xiphoid process and the opening of the peritoneum.

In a study of 1352 patients undergoing CABG with GEA grafting, the GEA patency rates were 97% at one month, 86% at five years and 67% at ten years [98]. Better long-term patency rates were reported when GEA was used as a skeletonized graft and after appropriate selection of target coronary vessel (>90% stenosis). In these settings, the patency rates were 98% at one month, 95% at five years and 90% at eight years [99]. Regarding target selection, a recent study identified the minimal coronary lumen diameter as a factor impacting GEA patency: when the lumen diameter was ≤1 mm, the patency rate was 90% at ten years compared to a patency rate of 40% for conduits anastomosed to vessels with diameter >1 mm [100].

## 6. Comparisons of Conduit Patency

There are few randomized trials that have directly compared the angiographic outcomes of bypass conduits, in particular RA vs. RITA. The Radial Artery Patency and Clinical Outcomes (RAPCO) system encompassed two randomized clinical trials comparing clinical and patency outcomes of the RA vs. free RITA (394 patients) and of the RA vs. SVG (225 patients), respectively [54]. With excellent completeness of follow-up (99%), the ten-year follow-up of the RAPCO demonstrated that the patency of RA was significantly higher than that of free RITA (89% vs. 80%; HR 0.45 95% CI 0.23–0.88), while the patency of RA was non-significantly better than that of the SVG (85% vs. 71%; HR 0.40 95% CI 0.15–1.00). Compared to free RITA, patients receiving RA also showed better long-term survival (91% vs. 84%, log-rank *p* = 0.03), while it was similar between RA and SVG (73% vs. 65%; log-rank *p* = 0.18).

The Radial Artery Database International Alliance (RADIAL) project compared the clinical and angiographic outcomes of the RA vs. SVG by pooling individual patient data from six trials [101]. In a pooled cohort of 1036 patients, the RADIAL investigators found that the use of RA was associated with a lower incidence of the composite endpoint of death, MI, and repeat revascularization (13% vs. 19%; HR 0.67, 95% CI 0.49–0.90) at a mean follow-up of 60 months, and with a lower incidence of graft occlusion (8% vs. 20%; HR 0.44, 95% CI 0.28–0.70) at a mean angiographic follow-up of 50 months.

In a recent network meta-analysis of 18 studies for a total of 8272 grafts analyzed, only the RA and the NT-SVG were found to be associated with a significantly lower graft occlusion rate when compared to conventionally harvested SVG at a mean weighted angiographic follow-up of five years [102]. The pooled patency rate was 94.1% (95% CI 90.0–97.6 for the RA, 91.4% (95% CI 87.3–94.3) for the NT-SVG, 89.2% (95% CI 71.2–96.5) for the RITA,86.3% (95% CI 81.2–90.2) for the conventionally harvested SVG and 61.2% (95% CI 52.2–69.4) for the GEA (Table 1).

## 7. Surgical Revascularization Technique and Graft Patency

### 7.1. On- and Off-Pump CABG

Off-pump CABG (OPCABG) is a key revascularization strategy that prevents patients from being exposed to the detrimental effects of cardio-pulmonary bypass and is currently recommended for patients with severe atherosclerotic disease of the aorta [2]. In the last decade, considerable randomized evidence has been published on both the clinical and angiographic outcomes of OPCABG compared to on-pump CABG (ONCABG).

In the Danish On-pump Versus Off-pump Randomization Study (DOORS), a total of 900 patients aged more than 70 years were assigned to either OPCABG or ONCABG and received the same heparinization protocol [103]. At six months, 481 patients underwent coronary angiography, and the proportion of occluded grafts was statistically higher in the OPCABG group vs. the ONCABG group (76% vs. 89%; *p* = 0.01).

Similarly, the Department of Veterans Affairs Randomized On/Off Bypass (ROOBY) trial, which randomized 2203 patients to OPCABG or ONCABG, found a statistically significant increased patency rate in ONCABG for both the arterial (91.4% vs. 85.8%; *p* = 0.003) and vein grafts (80.4% vs. 72.7%; *p* < 0.001) at 1 year [104].

The CABG Off or On Pump Revascularization Study (CORONARY) trial randomized 4752 patients to OPCABG or ONCABG [105]. In the smaller angiographic cohort, a total of 157 patients undergoing computed tomography angiography at one year were included. The investigators found that the patency index, that is, the proportion of nonoccluded grafts, was 89% in the OPCABG and 95% in the ONCABG (*p* = 0.09). Moreover, no difference in patency rates was found between arterial and venous grafts and target territories.

Recently, a meta-analysis of 16 RCTs totaling 6227 patients and 11,641 grafts found that OPCABG was associated with a 31% higher risk of graft failure after OPCABG (RR 1.31, 95% CI 1.17–1.46; *p* < 0.001) [106]. Notably, the higher graft occlusion was driven by studies in which the crossover rate, assumed as a proxy of surgeon expertise, was >10% (RR 1.31, 95% CI 1.16–1.49, *p* < 0.001), and there was no statistical difference in patency in studies with crossover rate <10%. Moreover, the occlusion rate in OPCABG was higher compared to ONCABG within the first year of follow-up (RR 1.34, 95% CI 1.18–1.52; *p* < 0.001) but was similar at longer follow-up (RR 1.15, 95% CI 0.87–1.52; *p* = 0.32), suggesting an effect from technical issues during the surgery. When focusing on the type of graft, SVGs were found to be more likely to fail in the OPCABG group (RR 1.40, 95% CI 1.23–1.59; *p* < 0.001), while no difference was found in the patency rates of arterial grafts between the two strategies.

Due to concerns related to graft patency, OPCABG is currently recommended in selected patients with severe aortic atherosclerosis and should be performed by experienced off-pump teams [1,2,107].

### 7.2. Minimally Invasive and Robotic CABG

In the last decade, more and more attention has been focused on reducing the invasiveness of cardiac surgical procedures paving the way to minimally invasive CABG and robotic CABG.

Minimally invasive direct CABG (MIDCAB) involves the combination of OPCABG and a minimally invasive approach through smaller surgical incisions, such as left anterolateral thoracotomy, reducing the risk of complications related to cardio-pulmonary bypass use and full sternotomy. Data on the patency of grafts after MIDCAB are scarce, and available results are mainly focused on the early patency rate of grafts. The patency rates for the LITA-LAD graft in the immediate postoperative period ranges from 96.2% to 99% [108,109,110], while at six months, it is reported to be between 95–100% for the LITA-LAD and 85% for the SVGs [109,111]. In one of the biggest series on MIDCAB, 1060 patients were follow-up for a median time of 11 years, and the patency rate of LITA-LAD was 96.8% [112]. Favorable results were also reported in a series of 140 MIDCAB patients in which the patency rate of the LITA-LAD at three years was 96.4% [113].

Robotic-assisted CABG (RCABG) represents a novel adjunct to reduce the invasiveness of cardiac surgery and achieve earlier functional recovery and lower morbidity. The reported patency rates after RCABG are similar to the rates of grafts used in conventional CABG. In a systematic review of the literature, which included 33 articles and a total of 4000 patients, the patency rates after RCABG were 97.7% at one month, 96.1% at one to five years and 93.2% after five years [114].

The optimal patency rates of minimally invasive techniques facilitate the role of these strategies in the current practice for selected patients assuring clinical and surgical benefits without compromising graft patency. An important caveat to these benefits is the expertise of the centers and surgeons performing such operations.

## 8. Angiographic and Clinical Outcomes

Excellent patency rates are key to achieve the clinical benefits of CABG. Despite the intuitive and biologically plausible relationship between graft status and clinical outcomes not being cleared yet, patent grafts are known to exert protective effects on the myocardium. Functioning grafts are able to not only allow flow distally to the coronary stenosis but also able to protect the proximal segment by preventing CAD progression [115].

The association between angiographic and clinical outcomes does not seem to be linear, and the consequences of graft failure are variable. For instance, failure of grafts to the LAD was reported to be more closely associated with worse clinical events than the failure of grafts to other territories. In an analysis of 1296 patients, clinical outcomes were similar between patients with and without SVGs stenosis at five years. However, significant stenosis of the SVGs grafted to the LAD was associated with a higher risk of death and cardiovascular events [116].

In the PREVENT-IV trial, SVG failure was associated with a higher risk of coronary revascularization but not death. However, failure of the LITA-LAD was strongly associated with a higher incidence of an acute clinical event [117].

In a provincial registry of 5276 patients, arterial graft failure, mainly LITA-LAD, was associated with lower survival compared to patients with vein graft failure [118].

Investigating the interplay between graft status and clinical outcomes is complex and influenced by several factors [119]. The lack of systematic angiographic graft assessment in most studies, which instead report on graft status following clinically driven angiographies, can bias the estimation of the actual impact of graft failure on clinical outcomes. Also, the temporal relationship between the two events is difficult to establish, provided that graft patency studies only determine if the graft failed and not when it occurred. It is essential to establish that graft failure occurred before the clinical event. The functionality of the failed graft should also be considered. If a graft is anastomosed to a coronary artery with a non-flow limiting lesion, competitive flow from the native coronary circulation will likely determine graft failure. However, this graft failure is likely less clinically relevant as no reduction in distal perfusion will occur, considering that native circulation will keep supplying the territory. Indeed, in the IMPAG (Impact of Preoperative FFR on Arterial Bypass Graft Function) [78] and SYNTAX-LE MANS (Synergy between PCI with TAXUS express and cardiac surgery left main angiographic substudy) [120] in which grafts either failed for competitive flow or were grafted in a situation of high risk of competitive flow (left main disease), no correlation was found between graft failure and clinical events.

Therefore, the impact of graft failure on patients’ prognosis is complex and highly variable and depends on the type and location of the failed graft and on the mechanism of failure.

## 9. Tailoring Surgical Revascularization to Patients

To achieve the best outcome for every patient, CABG and, in particular, the selection of conduits should be tailored to each patient.

The debate around the best second arterial graft is still ongoing in the surgical community. The current North American guidelines, however, indicate the RA as the second arterial conduit of choice to bypass highly stenosed coronary arteries [1].

With the available evidence, it is not possible to precisely define the CABG strategy for each patient. A complex interplay between patient characteristics (e.g., age, coronary anatomy, conduit availability), preoperative risk factors, patient expectations and surgeon expertise should be taken into consideration.

Life expectancy is supposed to play a major role in the decision-making process about using arterial or venous conduits. In the early follow-up period (<4 years), there is no difference in patency rates between arterial and venous grafts, but the patency rates start to be superior in arterial grafts four years after CABG [101,121]. This evidence suggests that patients with low life-expectancy would not benefit from the use of a second arterial graft. The lack of benefits for elderly patients is supported by some reports. In a post-hoc analysis of the ART, the use of BITA vs. SITA was associated with a lower risk of adverse cardiovascular events in younger patients but not in older patients [122]. A study including 26,124 patients identified an age cut-off of 70 years, after which the survival benefit of multiple arterial grafting is lost [123]. Consistently with this, the current European and North American guidelines recommend considering the use of multiple arterial grafts in patients with reasonable life expectancy [1,2].

Coronary anatomy plays a key role in the choice of conduits. As mentioned in the dedicated paragraphs, RA provides the highest benefit when used to bypass a severely stenosed coronary artery.

Patient comorbidities should guide conduit choice as well. Diabetes and obesity can represent important risk factors, particularly when concomitant, for the development of sternal wound complications in patients in which both ITAs are harvested. Therefore, these patients could benefit from other surgical strategies considering that sternal wound complications were reported to be associated with worse long-term outcomes [124].

Finally, the achievement of excellent outcomes after CABG also relies on surgeon and center expertise, as not every surgeon could be skillful in every procedure (multiple arterial grafting, off-pump, no-touch aortic technique) [125,126,127].

A proposed algorithm for the choice of the second conduit is shown in Figure 5.

The results from the ongoing Randomized Comparison of the Clinical Outcome of Single Versus Multiple Arterial Grafts (ROMA) trial will provide solid answers to these questions [128].

**Figure 5 jcm-12-02022-f005:**
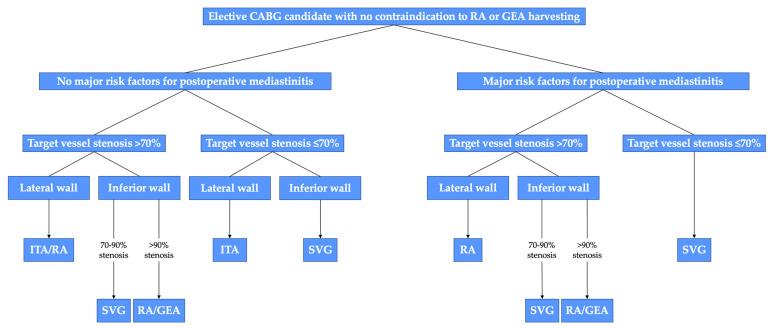
Algorithm for graft selection for the second target vessel in elective coronary artery bypass grafting (CABG) patients with stable coronary artery disease and without contraindications to the radial artery (RA) or gastroepiploic artery (GEA) harvesting. In case of contraindications to RA or GEA, SVG should be used. Risk factors for postoperative mediastinitis are obesity, diabetes, and severe chronic lung disease, especially in combination. ITA, internal thoracic artery; SVG, saphenous vein graft. Reproduced without changes from: “The choice of conduits in coronary artery bypass surgery” by Gaudino et al.; Journal of The American College of Cardiology. © 2015 American College of Cardiology Foundation. Published by Elsevier Inc. [129].

## 10. Conclusions

The benefits of surgical revascularization are best achieved with a cautious choice of bypass conduits. The LITA to LAD is the standard of care given the proven early and late benefits. However, the choice of the second conduit is still debated. Current evidence seems to consistently points towards a clinical and angiographic benefit from the RA compared to other conduits. Selecting the conduit with the best long-term patency is most likely to ensure optimal clinical outcomes in each patient if long-term patency is considered together with a tailored treatment strategy that accounts for patient characteristics (e.g., coronary anatomy, conduit availability), preoperative risks, patient expectations, and surgeon expertise.

## Figures and Tables

**Figure 1 jcm-12-02022-f001:**
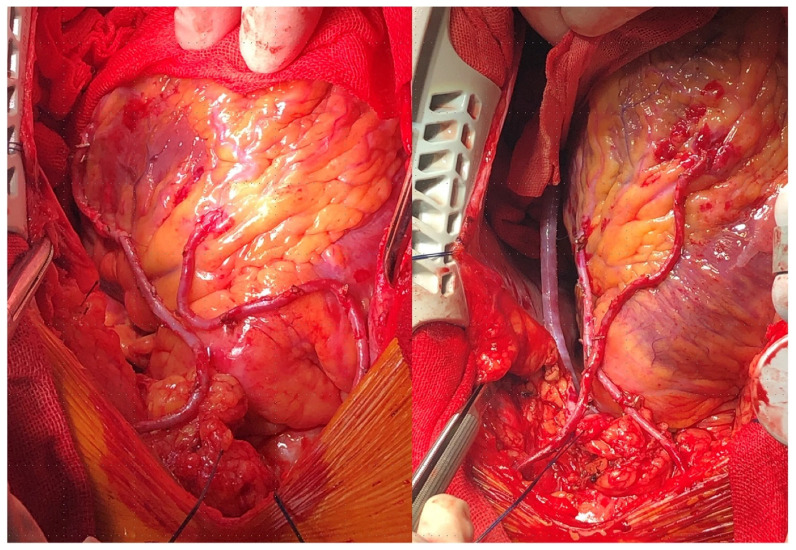
(**Left**) panel: In-situ right internal thoracic artery to left anterior descending artery with in-situ left internal thoracic artery to the diagonal branch. (**Right**) panel: In-situ right internal thoracic artery to left circumflex branch with in-situ left internal thoracic artery to left anterior descending. (Reproduced without changes from: “Application of bilateral internal mammary artery with different configurations in coronary artery bypass grafting” by Han et al.; Journal of Cardiothoracic Surgery, 2021; License: CC BY 4.0, https://creativecommons.org/licenses/by/4.0/, accessed on 15 January 2023).

**Figure 2 jcm-12-02022-f002:**
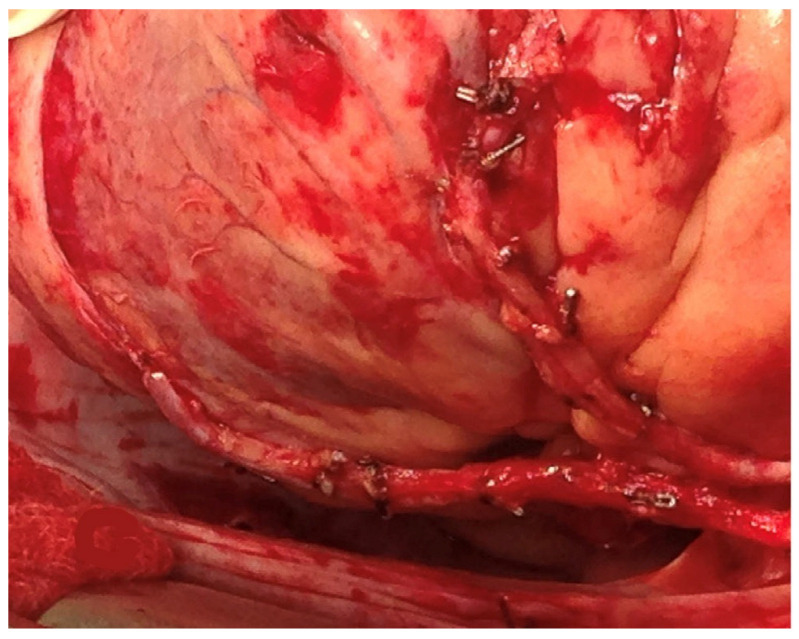
Left internal thoracic artery sequential grafting to the left anterior descending and diagonal arteries and right internal thoracic artery to the obtuse marginal artery. Reproduced from: “Hemodynamic changes during heart displacement in aorta no-touch off-pump coronary artery bypass surgery: a pilot study” by Carvalho et al.; Brazilian Journal of Cardio-Vascular Surgery, 2018; License: CC BY 4.0, https://creativecommons.org/licenses/by/4.0/, accessed on 20 February 2023).

**Figure 3 jcm-12-02022-f003:**
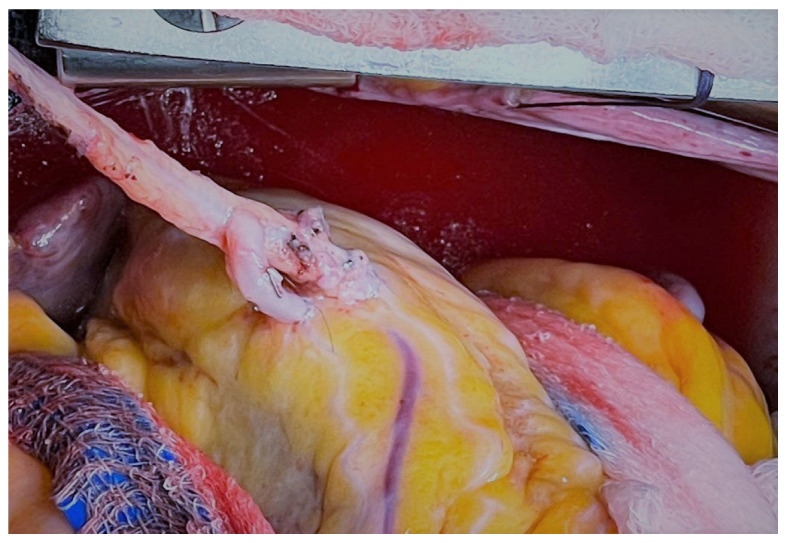
Baby-Y graft of radial artery and saphenous vein to obtuse marginal branches of left circumflex artery.

**Figure 4 jcm-12-02022-f004:**
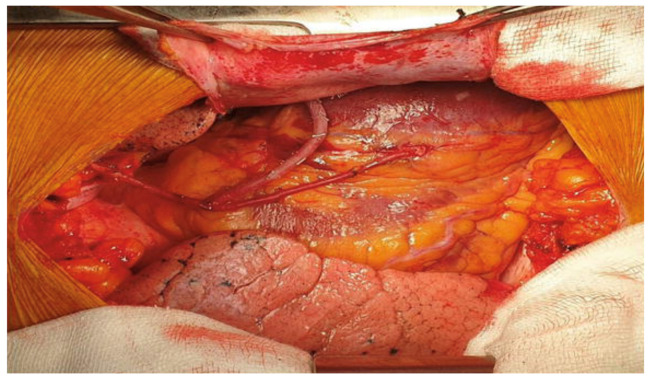
Left internal thoracic artery to the left anterior descending coronary artery and radial artery Y graft to diagonal artery. Reproduced without changes from: “Revascularization strategies in OPCABG” by Vettath et al.; Cardiac Surgery Procedures, 2019; License: CC BY 3.0, https://creativecommons.org/licenses/by/3.0/, accessed on 20 February 2023).

**Table 1 jcm-12-02022-t001:** Angiographic outcomes of currently used coronary artery bypass grafts.

Conduit	Patency Rate
RA	94.1% (95% CI 90.0–97.6)
No touch-SVG	91.4% (95% CI 87.3–94.3)
RITA	89.2% (95% CI 71.2–96.5)
Conventionally harvested SVG	86.3% (95% CI 81.2–90.2)
GEA	61.2% (95% CI 52.2–69.4)

Based on pooled estimates from a meta-analysis of 18 randomized clinical trials (8272 grafts) [102]. CI, confidence interval; LITA, left internal thoracic artery; GEA, gastroepiploic artery; RA, radial artery; RITA, right internal thoracic artery; SVG, saphenous vein graft.

## Data Availability

Not applicable.

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
