# Peer review of "Angiographic Outcomes for Arterial and Venous Conduits Used in CABG"

_jcm, 2023, doi:10.3390/jcm12052022_

Round 1

Reviewer 1 Report

Strengths of the Review

Thank you for the opportunity to review this manuscript. This review article addresses an important issue of current relevance: How to further improve short term and long-term results of CABG. The time of this debate is well chosen and the manuscript merits priority for publication. On the one side coronary surgery is obviously in a retreat situation, faced with the rapid progress of the various techniques of interventional cardiology and the medical prevention and treatment of coronary artery disease. On the other side important studies comparing surgical therapy versus PCI respectively medical treatment still indicate a benefit of coronary surgery in terms of improving outcome and prevention of myocardial infarction in the long-term. And recently new concepts of surgical revascularisation were adopted including minimal-invasive CABG (MICs-CABG) and hybrid approaches - the latter combining MICs-CABG with PCI. These techniques lower the patient`s threshold for the acceptance of surgical intervention but requires often trade-off between completeness of revascularisation, type and arrangement of bypass grafts.

The challenges in reviewing the strategies and concepts now lie so far in place, to identify the strengths of CABG and further promoting them.

In that sense the present article examines various graft materials, harvesting techniques and arrangements of bypass grafts at a glance and how CABG outcomes are influenced. The authors want to stand out from previous reviews about bypass grafts by selecting studies with primary focus on the patency of venous and arterial grafts using angiographic control rather than clinical outcomes.  In doing so, they sharpen and differentiate the biological properties of the venous and arterial vessels and its suitability as bypass graft. Notably, the choice of revascularization strategy premisses the individual patient demands based on the knowledge of the bypass graft characteristics.

The authors quoted the most important angiographic control studies at the expense of studies correlating the type of bypass graft and clinical outcome after CABG.  Another positive aspect of the review is, that they covered all important aspects of conduits used in CABG.  Based on relevant studies, it becomes clear that not only the type of graft itself – internal mammary artery, radial artery or venous conduits – influences the outcome, but its handling is potentially just as important: concerning the vein grafts, it’s durability may be improved by new methods of harvesting, where the surrounding tissue is preserved (No-Touch Technique) and optimized   pharmacotherapy (Statins and Double Platelet Inhibition).  Arterial grafts are more resistant to atherosclerosis and degeneration, but especially the radial artery becomes easily dysfunctional, when the stenosis of the corresponding coronary artery is not tight enough, resulting in increased flow resistant due to the competitive flows through the native coronaries. The authors quoted the appropriate studies, showing how graft patency rates may be optimized by applying individual graft configuration adapted to the coronary anatomy, amount of residual flow and severity of CAD.  Customized decision includes aortic versus composite grafts, sequential grafting versus single grafts, single versus multiple inflows.  Another positive aspect of this review that it contemporary reflects the debate of surgical innovation: e.g. the attempt to improve the patency of vein grafts by using external stents and the pros and cons of endoscopic vein harvesting.

Weaknesses of the review

When reading the review, it soon becomes clear that the evidence concerning type and handling of bypass grafts is still low, and that the results of the different studies are somewhat confusing.  For me the question raises what is the take-home message. The conclusion is formulated subsequently vaguely and in very broad terms.

At the beginning of the review the authors hint at a a problem, when stating that „current national registries in 27 the United States, United Kingdom, and globally report a low use of arterial grafts in patients undergoing CABG „ However, after reading this article readers expect a clear statement. They would wish to know more precisely weather and to what extend the patients prognosis is affected by type of revascularisation.  

Because J Clin Med targets mainly clinicians, who are not deeply involved in debates about choice of revascularisation strategies, but are interested in practical advices for their patients, the authors should spend more effort to better summarize the most important information in order to give an idea which revascularisation strategy is most suitable for which patient. The summary of the patency rates for arterial and venous grafts as provided in table 1 might be insufficient to complete the picture. For example, it is not clear at which time intervals differences in patency rates between the types of bypass grafts may get important. In this context I strongly encourage the authors to use better visual presentation of the decision algorithms that can be derived from this review. Flow charts are good option. Further the photographs are randomly combined instead of illustrating the main statements of the review. Creating photographs of selected revascularisation arrangements for the 3 or 4 most important patient’s prototypes would be helpful.

Author Response

Dear Prof. Royse,

Please find enclosed our revised manuscript entitled “Angiographic outcomes for arterial and venous conduits used in CABG” to be considered for publication in the Special Issue “Advances in Coronary Surgery Using Arterial Conduits” in the Journal of Clinical Medicine.

We have now revised our manuscript and amended it according to the insightful comments from the Reviewers. In particular, we have now described in detail the patency rates of sequential grafts, the patency rates following off-pump and minimally invasive CABG, and the relationship between angiographic and clinical outcomes. We have also provided further figures to better provide a graphical representation of our statements. Please find below a point-by-point reply to all comments.

Hoping that you would judge our manuscript of interest for the readers of Journal of Clinical Medicine, we thank you for your attention and look forward to hearing from you.

Sincerely,

Mario Gaudino, MD, MSCE, PhD

Stephen and Suzanne Weiss Professor in Cardiothoracic Surgery (II)

Assistant Dean for Clinical Trials

Director of the Joint Clinical Trials Office

Director of Translational and Clinical Research

Department of Cardiothoracic Surgery, Weill Cornell Medical Center

525 E 68th Street, New York, NY 10065
[email protected]

REVIEWER #1

Thank you for the opportunity to review this manuscript. This review article addresses an important issue of current relevance: How to further improve short term and long-term results of CABG. The time of this debate is well chosen and the manuscript merits priority for publication. On the one side coronary surgery is obviously in a retreat situation, faced with the rapid progress of the various techniques of interventional cardiology and the medical prevention and treatment of coronary artery disease. On the other side important studies comparing surgical therapy versus PCI respectively medical treatment still indicate a benefit of coronary surgery in terms of improving outcome and prevention of myocardial infarction in the long-term. And recently new concepts of surgical revascularisation were adopted including minimal-invasive CABG (MICs-CABG) and hybrid approaches - the latter combining MICs-CABG with PCI. These techniques lower the patient`s threshold for the acceptance of surgical intervention but requires often trade-off between completeness of revascularisation, type and arrangement of bypass grafts.

The challenges in reviewing the strategies and concepts now lie so far in place, to identify the strengths of CABG and further promoting them.

In that sense the present article examines various graft materials, harvesting techniques and arrangements of bypass grafts at a glance and how CABG outcomes are influenced. The authors want to stand out from previous reviews about bypass grafts by selecting studies with primary focus on the patency of venous and arterial grafts using angiographic control rather than clinical outcomes.  In doing so, they sharpen and differentiate the biological properties of the venous and arterial vessels and its suitability as bypass graft. Notably, the choice of revascularization strategy premisses the individual patient demands based on the knowledge of the bypass graft characteristics.

The authors quoted the most important angiographic control studies at the expense of studies correlating the type of bypass graft and clinical outcome after CABG.  Another positive aspect of the review is, that they covered all important aspects of conduits used in CABG.  Based on relevant studies, it becomes clear that not only the type of graft itself – internal mammary artery, radial artery or venous conduits – influences the outcome, but its handling is potentially just as important: concerning the vein grafts, it’s durability may be improved by new methods of harvesting, where the surrounding tissue is preserved (No-Touch Technique) and optimized   pharmacotherapy (Statins and Double Platelet Inhibition).  Arterial grafts are more resistant to atherosclerosis and degeneration, but especially the radial artery becomes easily dysfunctional, when the stenosis of the corresponding coronary artery is not tight enough, resulting in increased flow resistant due to the competitive flows through the native coronaries. The authors quoted the appropriate studies, showing how graft patency rates may be optimized by applying individual graft configuration adapted to the coronary anatomy, amount of residual flow and severity of CAD.  Customized decision includes aortic versus composite grafts, sequential grafting versus single grafts, single versus multiple inflows.  Another positive aspect of this review that it contemporary reflects the debate of surgical innovation: e.g. the attempt to improve the patency of vein grafts by using external stents and the pros and cons of endoscopic vein harvesting.

 Reply: thank you for your thorough and supportive comment.

Comment #1

When reading the review, it soon becomes clear that the evidence concerning type and handling of bypass grafts is still low, and that the results of the different studies are somewhat confusing.  For me the question raises what is the take-home message. The conclusion is formulated subsequently vaguely and in very broad terms. At the beginning of the review the authors hint at a problem, when stating that „current national registries in 27 the United States, United Kingdom, and globally report a low use of arterial grafts in patients undergoing CABG „ However, after reading this article readers expect a clear statement. They would wish to know more precisely weather and to what extend the patients prognosis is affected by type of revascularisation.  

Reply: thank you. We have now added a paragraph which describes the relationship between graft patency and patient prognosis. This paragraph adds to the clinical correlations already provided in the paragraphs dedicated to the conduits.

Change:

Lines 591-629:

 “8. Angiographic and clinical outcomes

Excellent patency rates are key to achieve the clinical benefits of CABG. Despite the intuitive and biologically plausible relationship between graft status and clinical outcomes has not been cleared yet, patent grafts are known to exert protective effects on the myocardium. Functioning grafts are able to not only allow flow distally to the coronary stenosis but are also able to protect the proximal segment by preventing CAD progression [114].

The association of angiographic and clinical outcome does not seem to be linear, and consequences of graft failure are variable. For instance, failure of grafts to the LAD was reported to be more closely associated with worse clinical events than failure of grafts to other territories. In an analysis of 1,296 patients’ clinical outcomes were similar between patients with and without SVGs stenosis at 5 years, however significant stenosis of the SVGs grafted to the LAD was associated with a higher risk of death and cardiovascular events [115].

In the PREVENT-IV trial, SVG failure was associated with a higher risk of coronary revascularization but not death, however failure of the LITA-LAD was strongly associated with a higher incidence of acute clinical event [116].

In a provincial registry of 5,276 patients, arterial graft failure, mainly LITA-LAD, was associated with lower survival compared to patients with vein graft failure [117].

Investigating the interplay between graft status and clinical outcomes is complex and influenced by several factors [118]. The lack of systematic angiographic graft assessment in most studies, which instead report on graft status following clinically driven angiographies, can bias the estimation of the actual impact of graft failure on clinical outcomes. Also, the temporal relationship between the two events is difficult to establish provided that graft patency studies only determine if the graft failed and not when it occurred. It is essential to establish that graft failure occurred before the clinical event [118]. The functionality of the failed graft should also be considered. If a graft is anastomosed to a coronary artery with a non-flow limiting lesion, competitive flow from the native coronary circulation will likely determine graft failure. However, this graft failure is likely less clinically relevant as no reduction in distal perfusion will occur considering that native circulation will keep supplying the territory. Indeed, in the IMPAG (Impact of Pre-operative FFR on Arterial Bypass Graft Function) [119] and SYNTAX-LE MANS (Synergy between PCI with TAXUS express and cardiac surgery left main angiographic sub study) [120] in which grafts either failed for competitive flow or were grafted in a situation of high risk of competitive flow (left main disease), no correlation was found between graft failure and clinical events.

Therefore, the impact of graft failure on patients’ prognosis is complex and highly variables and depends on the type and location of the failed graft and on the mechanism of failure.”

Comment #2

Because J Clin Med targets mainly clinicians, who are not deeply involved in debates about choice of revascularisation strategies, but are interested in practical advices for their patients, the authors should spend more effort to better summarize the most important information in order to give an idea which revascularisation strategy is most suitable for which patient. The summary of the patency rates for arterial and venous grafts as provided in table 1 might be insufficient to complete the picture. For example, it is not clear at which time intervals differences in patency rates between the types of bypass grafts may get important.

Reply: A paragraph dedicated to tailoring surgical revascularization strategies has now been added to provide more insights on which characteristics can make a conduit more suitable and beneficial to a patient.

Changes:

Lines 631-667:

9. Tailoring surgical revascularization to patients

To achieve the best outcome in every patient, CABG and, in particular, selection of conduits, should be tailored to each patient.

The debate around the best second arterial graft is still ongoing in the surgical community. The current North American guidelines, however, indicate the RA as the second arterial conduit of choice to bypass highly stenosed coronary arteries[1][121].

With the available evidence, it is not possible to precisely define the CABG strategy for each patient. A complex interplay between patient characteristics (e.g., age, coronary anatomy, conduit availability), preoperative risk factors, patient expectations and surgeon expertise should be taken into consideration.

Life expectancy is supposed to play a major role in the decision-making process about using arterial or venous conduits. In the early follow-up period (<4 years) there is no difference in patency rates between arterial and venous grafts, but the patency rates start to be superior in arterial graft 4 years after CABG[100,121]. This evidence suggests that patients with low life-expectancy would not benefit from the use of a second arterial graft. The lack of benefit in elderly patients is supported by some reports. In a post-hoc analysis of the ART the use of BITA vs SITA was associated with a lower risk of adverse cardio-vascular events in younger patients but not in older patients[122]. A study including 26,124 patients identified an age cut-off of 70 years after which the survival benefit of multiple arterial grafting is lost[123]. Consistently with this, the current European and North American guidelines recommend considering the use of multiple arterial grafts in patients with reasonable life expectancy[1,2].

Coronary anatomy plays a key role in the choice of conduits. As mentioned in the dedicated paragraphs, RA provides the highest benefit when used to bypass a severely stenosed coronary artery.

Patient comorbidities should guide conduit choice as well. Diabetes and obesity can represent important risk factors, particularly when concomitant, for the development of sternal wound complications in patients in which both ITAs are harvested and therefore these patients could benefit from other surgical strategies considering that sternal wound complications were reported to be associated with worse long-term outcomes[124].

Finally, achievement of excellent outcomes after CABG also relies on surgeon and center expertise, as not every surgeon could be skillful in every procedure (multiple arterial grafting, off-pump, no-touch aortic technique)[125–127].

A proposed algorithm for the choice of the second conduit is shown in Figure 5.

The results from the ongoing Randomized Comparison of the Clinical Outcome of Single Versus Multiple Arterial Grafts (ROMA) trial will provide solid answers to these questions[128].

Comment #3

In this context I strongly encourage the authors to use better visual presentation of the decision algorithms that can be derived from this review. Flow charts are good option.

Reply: thank you. We have now added a flowchart which depicts the algorithm for the choice of the second conduit.

Changes: see Figure 5.

Comment #4

Further the photographs are randomly combined instead of illustrating the main statements of the review. Creating photographs of selected revascularisation arrangements for the 3 or 4 most important patient’s prototypes would be helpful.

Reply: We have now added more figures to better provide a graphical representation of our statements.

Changes: see Figure 1, 2, 3 and 4. 

Reviewer 2 Report

Authors have reported a systematic review on conduits used in CABG surgery with the aim of provide the current evidence on patency of these grafts.

The paper is interesting and well-written, covering a lot of aspects regarding each type of conduit generally used in coronary surgery. Some considerations:

-       Authors should provide a separate section to the comparison between on-pump and beating-heart CABG. A lot of literature has emerged over last years about graft patency in these different surgical techniques, and a discussion of this topic should be addressed in this review.

-       Sequential anastomoses are generally used during CABG surgery; authors have focused their attention only on sequential grafting using radial artery. What about ITA of SV grafts? It could be very interesting to know the evidence of literature regarding the rate of patency of individual and sequential anastomoses using these grafts, that are the most used in the cardiac surgery community.

-       Sequential grafting with the ITA graft could require the use of two different anastomotic techniques: a diamond anastomosis (constructed with the conduit perpendicular to the coronary target) or a parallel anastomosis (constructed when the conduit is parallel to the coronary target). Is there any evidence about differences in graft patency with these different techniques?

-       Finally, nowadays an increasing interest is directed to minimally invasive techniques. MIDCAB (with direct ITA harvesting) and robotic CABG are emerged as alternatives. A separate section that pays attention on the effect of the surgical approach on the graft patency should be added.

Author Response

Dear Prof. Royse,

Please find enclosed our revised manuscript entitled “Angiographic outcomes for arterial and venous conduits used in CABG” to be considered for publication in the Special Issue “Advances in Coronary Surgery Using Arterial Conduits” in the Journal of Clinical Medicine.

We have now revised our manuscript and amended it according to the insightful comments from the Reviewers. In particular, we have now described in detail the patency rates of sequential grafts, the patency rates following off-pump and minimally invasive CABG, and the relationship between angiographic and clinical outcomes. We have also provided further figures to better provide a graphical representation of our statements. Please find below a point-by-point reply to all comments.

Hoping that you would judge our manuscript of interest for the readers of Journal of Clinical Medicine, we thank you for your attention and look forward to hearing from you.

Sincerely,

Mario Gaudino, MD, MSCE, PhD

Stephen and Suzanne Weiss Professor in Cardiothoracic Surgery (II)

Assistant Dean for Clinical Trials

Director of the Joint Clinical Trials Office

Director of Translational and Clinical Research

Department of Cardiothoracic Surgery, Weill Cornell Medical Center

525 E 68th Street, New York, NY 10065
[email protected]

REVIEWER #2

Authors have reported a systematic review on conduits used in CABG surgery with the aim of provide the current evidence on patency of these grafts.

The paper is interesting and well-written, covering a lot of aspects regarding each type of conduit generally used in coronary surgery. Some considerations:

Reply: thank you for your comment.

Comment #1

Authors should provide a separate section to the comparison between on-pump and beating-heart CABG. A lot of literature has emerged over last years about graft patency in these different surgical techniques, and a discussion of this topic should be addressed in this review.

Reply: thank you. A paragraph on patency rate after on-pump and off-pump CABG has been added.

Change:

Lines 528-564:

On- vs off-pump CABG

Off-pump CABG (OPCABG) is a key revascularization strategy which prevents the patients from being exposed to the detrimental effects of cardio-pulmonary bypass and is currently recommended for patients with severe atherosclerotic disease of the aorta [2][2]. In the last decade considerable randomized evidence has been published on both the clinical and angiographic outcomes of OPCABG compared to on-pump CABG (ONCABG).

In the Danish On-pump Versus Off-pump Randomization Study (DOORS) a total of 900 patients aged more than 70 years were assigned to either OPCABG or ONCABG and received the same heparinization protocol [101][100]. At 6 months, 481 patients underwent coronary angiography, and the proportion of occluded graft was statistically higher in the OPCABG group vs the ONCABG group (76% vs 89%; P=0.01).

Similarly, the Department of Veterans Affairs Randomized On/Off Bypass (ROOBY) trial, which randomized 2,203 patients to OPCABG or ONCABG, found a statistically significant increased patency rate in ONCABG for both the arterial (91.4% vs 85.8%; P=0.003) and vein grafts (80.4% vs 72.7%; P<0.001) at 1 year [102][101].

The CABG Off or On Pump Revascularization Study (CORONARY) trial randomized 4,752 patients to OPCABG or ONCABG [103][102]. In the smaller angiographic cohort, a total of 157 patients undergoing computed tomography angiography at 1 year were included. The investigators found that the patency index, that is the proportion of nonoccluded grafts, was 89% in the OPCABG and 95% in the ONCABG (P=0.09). Moreover, no difference in patency rates was found between arterial and venous grafts and target territories.

Recently, a meta-analysis of 16 RCTs totaling 6,227 patients and 11,641 grafts found that OPCABG was associated with a 31% higher risk of graft failure after OPCABG (RR 1.31, 95% CI 1.17-1.46; P<0.001) [104][103]. Notably, the higher graft occlusion was driven by studies in which cross-over rate, assumed as a proxy of surgeon expertise, was >10% (RR 1.31, 95% CI 1.16-1.49, P<0.001) and there was no statistical difference in patency in studies with cross-over rate <10%. Moreover, the occlusion rate in OPCABG was higher compared to ONCABG within the first year of follow-up (RR 1.34, 95% CI 1.18–1.52; P < 0.001) but was similar at longer follow-up (RR 1.15, 95% CI 0.87–1.52; P = 0.32), suggesting an effect from technical issues during the surgery. When focusing on type of graft, SVGs were found to be more likely to fail in the OPCABG group (RR 1.40, 95% CI 1.23–1.59; P < 0.001) while no difference was found in the patency rates of arterial grafts between the two strategies.

Due to concerns related to graft patency, OPCABG is currently recommended in selected patients with severe aortic atherosclerosis and should be performed by experienced off-pump teams [1,2,106].”

Comment #2

Sequential anastomoses are generally used during CABG surgery; authors have focused their attention only on sequential grafting using radial artery. What about ITA of SV grafts? It could be very interesting to know the evidence of literature regarding the rate of patency of individual and sequential anastomoses using these grafts, that are the most used in the cardiac surgery community.

Reply: A paragraph describing the patency rates for sequential ITA and a paragraph on patency rate of sequential vein grafts have been added to the manuscript.

Changes:

Lines 218-238

The impact of anastomotic technique

SVGs can be grafted as an individual anastomosis or as a sequential anastomosis, which allows for a greater extent of myocardial revascularization. However, there is limited data on the impact of sequential grafting on SVGs patency.

In a study of 2,515 patients, the clinical and angiographic outcomes in 946 sequential SVGs and 1,366 individual SVGs were compared [45]. The graft failure rates were 10.3% and 17.7% at 5 years and 20.9% and 33.6% at 10 years in the sequential and individual groups, respectively. After propensity score adjustment, the risk of graft failure was lower in the sequential group at a median follow-up of 88 months (HR 0.69, 95% CI 0.50-0.95; P=0.02). The risk of the composite outcome of death, nonfatal MI, and repeat revascularization was numerically lower in the sequential group (36.8% vs 41.4%; HR 0.91, 95% CI 0.75-1.09; P=0.30). Conversely, in a sub-study of Project of Ex-Vivo Vain Graft Engineering via Transfection (PREVENT) IV trial, which enrolled 3,014 patients to receive either edifoligide-treated SVGs or placebo, investigators found a higher risk of graft occlusion with sequential anastomosis at 1 year (OR 1.24, 95% CI 1.03-1.48; P=0.025) [46].

A recent meta-analysis including 15 cohort studies, totaling 10,681 patients and 12,957 grafts found that patency was higher for sequential SVGs compared to individual SVGs (relative risk [RR] 1.11, 95% CI 1.03–1.21; P=0.01) [47].

Due to conflicting results, more solid evidence, potentially from RCTs, is welcomed to explore the impact of individual vs sequential anastomosis on SVGs patency.”

Lines 372-404

The impact of anastomotic technique

The use of sequential anastomosis for ITA can be key to achieve effective revascularization in patients with a limited number of grafts and diffuse CAD (Figure 2). Current evidence, albeit based on observational studies, seems to be concordant with sequential ITA having a similar patency rate as individual ITA [74–77]. A recent study including 120 propensity score matched pairs reported that the patency rate of the sequential LITA graft to the diagonal artery and LAD was similar to the patency of individual LITA-to-LAD grafts (99% and 98% vs 98%; P>0.05) at the angiographic follow-up of 27 months. [77]. Similar rates were found in a report of 101 patients in which patency rates for the sequential LITA to the circumflex and right coronary artery were 98% and 95%, respectively [75].

Interestingly, the design of the sequential anastomosis was reported to impact graft patency [76]. In sequential ITA grafting, the anastomosis can be performed either in a “diamond” fashion with the conduit perpendicular to the coronary target or in a parallel fashion with the conduit parallel to the coronary target. In a study of 452 patients comparing LITA sequential vs individual grafting to the circumflex showed that a “diamond” anastomosis had a numerically higher patency rate of the distal segment of the sequential graft as compared to a parallel design (98.4% vs 90.7%; P=0.09). In particular, the lowest patency rate (75%) was reported when the proximal anastomosis of a sequential graft to the diagonal and circumflex arteries was placed in a parallel fashion [76].

However, it is important to note that the “diamond” design is more technically demanding and require accurate precision of the arteriotomies in order to prevent a “seagull effect”, which can lead to flattening of the graft and impairment of graft status.

Another important factor potentially affecting sequential graft success is coronary target stenosis. Lower stenosis can result in higher competitive flow and therefore jeopardize the functionality of the graft[78]. In the Impact of Preoperative FFR on Arterial Bypass Graft Function (IMPAG) trial which included a total of 64 patients (199 anastomosis, of whom 108 sequential), a preoperative fractional flow reserve (FFR) of at least 0.78 was associated with a better graft functionality at the 6-mont angiographic assessment. Lower values were associated with an anastomotic occlusion rate of 3%[79]. In a post hoc analysis of the IMPAG trial, specifically looking at sequential graft, a cut-off of 0.80 and 0.78 were found for the first and second anastomoses of sequential grafts to the anterior circulation compared to 0.74 for the individual grafts. The FFR cut-off were higher for grafts to the postero-lateral territory being 0.81 and 0.78 for the first and second anastomoses of sequential grafts and 0.79 for individual grafts[80]. These findings suggest that attention should be paid to the severity of the target coronary vessel when sequential anastomosis are planned.”

Comment #3

Sequential grafting with the ITA graft could require the use of two different anastomotic techniques: a diamond anastomosis (constructed with the conduit perpendicular to the coronary target) or a parallel anastomosis (constructed when the conduit is parallel to the coronary target). Is there any evidence about differences in graft patency with these different techniques?

Reply: thank you. We have now presented the evidence on the patency rate of the diamond and parallel anastomosis for the ITA sequential graft.

Changes:

Lines 388-399

Interestingly, the design of the sequential anastomosis was reported to impact graft patency [76]. In sequential ITA grafting, the anastomosis can be performed either in a “diamond” fashion with the conduit perpendicular to the coronary target or in a parallel fashion with the conduit parallel to the coronary target. In a study of 452 patients’ comparing LITA sequential vs individual grafting to the circumflex showed that a “diamond” anastomosis had a numerically higher patency rate of the distal segment of the sequential graft as compared to a parallel design (98.4% vs 90.7%; P=0.09). In particular, the lowest patency rate (75%) was reported when the proximal anastomosis of a sequential graft to the diagonal and circumflex arteries was placed in a parallel fashion [76].

However, it is important to note that the “diamond” design is more technically demanding and require accurate precision of the arteriotomies in order to prevent a “seagull effect”, which can lead to flattening of the graft and impairment of graft status.”

Comment #4

Finally, nowadays an increasing interest is directed to minimally invasive techniques. MIDCAB (with direct ITA harvesting) and robotic CABG are emerged as alternatives. A separate section that pays attention on the effect of the surgical approach on the graft patency should be added.

Reply: A discussion on minimally invasive techniques such as MIDCAB and robotic CABG has now been added to the manuscript.

Changes:

Lines 565-589

Minimally invasive and robotic CABG

In the last decade, more and more attention has been focused on reducing the invasiveness of cardiac surgical procedures paving the way to minimally invasive CABG and robotic CABG.

Minimally invasive direct CABG (MIDCAB) involve the combination of OPCABG and a minimally invasive approach through smaller surgical incisions, such as left anterolateral thoracotomy, reducing the risk of complications related to cardio-pulmonary bypass use and full sternotomy. Data on patency of grafts after MIDCAB are scarce and available results are mainly focused on the early patency rate of grafts. The patency rates for the LITA-LAD graft in the immediate postoperative period ranges from 96.2% to 99% [107–109] while at 6 months it is reported to be between 95%-100% for the LITA-LAD and 85% for the SVGs [108,110]. In one of the biggest series on MIDCAB, 1,060 patients were follow-up for a median time of 11 years and the patency rate of LITA-LAD was 96.8% [111]. Favorable results were also reported in a series of 140 MIDCAB patients in which patency rate of the LITA-LAD at 3 years was 96.4% [112].

Robotic-assisted CABG (RCABG) represents a novel adjunct to reduce invasiveness of cardiac surgery and achieve earlier functional recovery and lower morbidity. The reported patency rates after RCABG are similar to the rates of grafts used in conventional CABG. In a systematic review of the literature which included 33 articles and a total of 4,000 patients, the patency rates after RCABG were 97.7% at 1 month, 96.1% at 1-5 years and 93.2% after 5 years.[113].

The optimal patency rates of minimally invasive techniques facilitate the role of these strategies in the current practice for selected patients assuring clinical and surgical benefits without compromising graft patency. An important caveat to these benefits is the expertise of the centers and surgeons performing such operations.”